# The Th1 cell regulatory circuitry is largely conserved between human and mouse

Stephen Henderson[1], Venu Pullabhatla[2], Arnulf Hertweck[3], Emanuele de Rinaldis[2], Javier Herrero[1], Graham M Lord[2,4,5], Richard G Jenner[3]

**Gene expression programs controlled by lineage-determining transcription factors are often conserved between species. However, infectious diseases have exerted profound evolutionary pressure, and therefore the genes regulated by immune-specific transcription factors might be expected to exhibit greater divergence. T-bet (Tbx21) is the immune-specific, lineage-specifying transcription factor for T helper type I (Th1) immunity, which is fundamental for the immune response to intracellular pathogens but also underlies inflammatory diseases. We compared T-bet genomic targets between mouse and human CD4+ T cells and correlated T-bet binding patterns with species-specific gene expression. Remarkably, we found that the majority of T-bet target genes are conserved between mouse and human, either via preservation of binding sites or via alternative binding sites associated with transposon-linked insertion. Species-specific T-bet binding was associated with differences in transcription factor–binding motifs and species-specific expression of associated genes. These results provide a genome-wide cross-species comparison of Th1 gene regulation that will enable more accurate translation of genetic targets and therapeutics from pre-clinical models of inflammatory and infectious diseases and cancer into human clinical trials.**

## Introduction

The differentiation of naïve CD4+ T cells into T helper type 1 (Th1) effector cells tailors the immune response to target intracellular bacteria and viruses (1). However, inappropriate Th1 effector cell activation contributes to the development of autoimmune and inflammatory diseases.

The differentiation of naïve CD4+ T cells into Th1 effectors is controlled by the lineage-specifying transcription factor T-bet. Experiments in genetically modified mice have revealed that T-bet is necessary and sufficient for Th1 differentiation (2, 3). T-bet directly activates genes such as those encoding the inflammatory cytokines IFNγ and TNF and the receptors TIM3 (encoded by *HAVCR2*) and CCR5 (4, 5, 6, 7, 8, 9, 10, 11, 12, 13, 14). At these genes, T-bet binds to extended *cis*-regulatory regions (super-enhancers) (12, 15, 16) and recruits Mediator and P-TEFb to activate transcription (16). T-bet also interacts with the H3K4 methyltransferase SETD7 and the H3K27 demethylase KDM6B, recruiting these factors to *Ifng* (17). Genetic variation at T-bet binding sites is associated with differences in T-bet occupancy, including at causal variants associated with inflammatory disease (18), suggesting that differences in T-bet binding between individuals directly contributes to disease risk.

Much of our understanding of Th1 cell function in health and disease comes from studies in mice but the degree to which these findings can be applied to humans is unclear, especially given the evolutionary pressure on the immune system exerted by pathogens (19, 20, 21). Indeed, a recent study of a human patient homozygous for an inactivating T-bet mutation suggested a requirement for T-bet for immunity against mycobacteria in humans but not against other intracellular pathogens such as viruses (22). Comparison between the expression profiles of in vitro activated human and mouse T cells has revealed that the T-cell activation program is generally shared, but that significant differences do exist between the two species (19, 23). This variation may be due to differences in transcription factor binding. As an example of this, we have previously found that the gene encoding the colon homing receptor GPR15 is only occupied by GATA3 and expressed in human and not mouse Th2 cells (24). In mouse, expression of the gene is instead specific to Th17 cells and peripherally derived induced regulatory T cells (pT$_{REG}$), resulting in differences in the types of T helper cells that are trafficked to the human versus mouse colon. Thy-1 (CD90) is

[1]Bill Lyons Informatics Centre, UCL Cancer Institute and CRUK UCL Centre, University College London, London, UK   [2]NIHR Biomedical Research Centre at Guy's and St Thomas' Hospital and King's College London, London, UK   [3]Regulatory Genomics Group, UCL Cancer Institute and CRUK UCL Centre, University College London, London, UK   [4]School of Immunology and Microbial Sciences, King's College London, London, UK   [5]Faculty of Biology, Medicine and Health, University of Manchester, Manchester, UK

Correspondence: r.jenner@ucl.ac.uk;  graham.lord@manchester.ac.uk
Venu Pullabhatla's present address is Oxford Gene Technology, Oxford, UK
Emanuele de Rinaldis's present address is Sanofi, Cambridge, MA, USA
Arnulf Hertweck's present address is Nucleome Therapeutics Limited, Oxford, UK

used as a T-cell marker in mice but, in humans, it is only expressed on other cell types, potentially depending on the presence or absence of an Ets-1 binding site in the third intron of the gene (25). However, although differences in transcription factor occupancy have been compared systematically between humans and mice in other cell types, notably hepatocytes (26), such analyses have not been performed for immune cells. Thus, the similarities and differences between human and mouse Th1 cell transcriptional programs remain unknown.

# Results

## Identification of shared and species-specific T-bet binding sites

We sought to identify the degree to which T-bet binding sites and target genes were conserved between human and mouse Th1 cells. We gathered ChIP-seq data from human and mouse Th1 cells differentiated in vitro from purified CD4$^+$ T cells (12, 13, 14, 18) (Table S1). We first identified the genome positions consistently occupied by T-bet in each species at high confidence (q < 0.01 in all replicate ChIP-seq datasets; Fig S1A and Table S2) and then identified the subset of these regions that could be compared between species using liftOver (27) (which mapped the position of 90% of binding sites to the other species [Fig S1A]). Conserved binding sites were defined as those bound at high confidence in both species and species-specific sites as those bound at high confidence in one species and for which there was no evidence of binding in the other species (q > 0.1 in all replicates). Binding sites outside of these criteria were judged as indeterminate and were not considered further. This process revealed that around one-third of T-bet binding sites were conserved between species (3,042 [36%] in humans; 2,749 [32%] in mouse) and around two-thirds of sites exhibited species-specific binding.

To compare T-bet gene targeting between human and mouse, we focused on T-bet binding sites associated with orthologous genes, which gave 2,191 genes occupied by T-bet in either species. At the majority (1,521, 69%) of these genes, a specific T-bet binding site was conserved between species (conserved target genes, Fig 1A and B and Table S3). These genes included the classical Th1 genes *IFNG*, *CXCR4*, *FASLG*, *HAVCR2*, *IL12RB2*, *IL18R1*, *IL18RAP*, and *TNF* (Figs 1C and S1B) (1, 2, 3, 4, 5, 6, 11, 12, 13, 14, 18, 28, 29, 30, 31, 32). An additional 349 genes (16%) were also bound by T-bet in both species but at alternative species-specific sites (alternative, Fig 1A and B and Table S3), including the genes *TNFSF15*, *CCR2*, and *MGAT4A* (Figs 1C and S1C). Finally, 171 genes (8%) were only bound by T-bet in humans (Hs-specific target genes) and 150 genes (7%) were only bound by T-bet in mouse (Mm-specific target genes; Table S3). Hs-specific T-bet target genes included *GREM2*, *TIMD4*, and *VCAM1*, whereas Mm-specific T-bet target genes included *Il18*, *Serpinb5*, and *Bend4* (Figs 1C and S1D and E). Thus, we can draw three conclusions from this analysis. First, the majority of T-bet target genes are conserved between mouse and human. Second, loss of a binding site at a gene in one species tends to be accompanied

by the appearance of an alternative site at the same gene in the other species. Finally, for only a relatively small number of genes is T-bet binding unique to human or mouse.

## Species-specific recruitment of transcriptional co-factors at T-bet binding sites

We next sought to address whether the species-specific T-bet binding sites we identified were likely to be functional. We have previously found that T-bet recruits P-TEFb, Mediator (MED1 sub-unit), and the super elongation complex (AFF4 subunit) to its binding sites in human and mouse Th1 cells (16). We therefore asked whether species-specific T-bet binding was accompanied by species-specific recruitment of these factors. We gathered ChIP-seq data for these transcriptional regulators in human and mouse Th1 cells and plotted the occupancy of the factors at conserved, alternative, and species-specific sites. We found that all of the factors were enriched at conserved sites in human and mouse, consistent with T-bet recruiting these factors in both species (Figs 2 and S2A–C). In contrast, species-specific T-bet binding sites were only occupied by P-TEFb, AFF4, and MED1 in the species in which T-bet was bound. This was also the case for genes bound by T-bet at alternative sites in humans and mouse, with the co-factors only occupying the sites at which T-bet was present in that species. This demonstrates that loss of T-bet binding is accompanied by a loss of activity of the associated regulatory element, and therefore other factors are not able to compensate for the loss of T-bet function at these sites. However, although T-bet occupancy was similar at alternative and species-specific sites, P-TEFb, AFF4, and MED1 occupancy was reduced compared with conserved sites, suggesting that T-bet may not exhibit the same functionality at these sites as it does at conserved sites. We conclude that species-specific T-bet binding results in species-specific recruitment of T-bet–dependent co-factors, although to a lesser extent compared with T-bet binding sites that are conserved between species.

## Species-specific T-bet binding is associated with species-specific gene expression

We next sought to determine whether these patterns of T-bet binding were associated with differences in gene expression between species. To avoid potential issues with differences in expression being dataset-dependent rather than species-dependent, we performed differential gene expression analysis between human and mouse using three independent in vitro polarised Th1 cell RNA-seq datasets for each species (16, 33, 34, 35, 36, 37) (Fig S3A and Tables S1 and S4. We found that genes associated with conserved binding sites exhibited similar expression levels in human and mouse (mean $\log_2$ human/mouse expression ratio of −0.59, std. dev 1.59; Fig 3A). Genes bound by T-bet in both species, but at alternative sites, exhibited more variable expression between species (std dev. 2.46, F 0.42, p < 2e$^{-16}$), but a similar mean $\log_2$ human/mouse expression ratio (−0.21). Thus, for many genes, the loss of T-bet binding during evolution may be functionally neutral if the binding site is replaced by an alternative T-bet binding site at the gene. In contrast, genes bound by T-bet specifically in human tended to be more highly expressed in human (mean $\log_2$ Hs/Mm of 1.95,

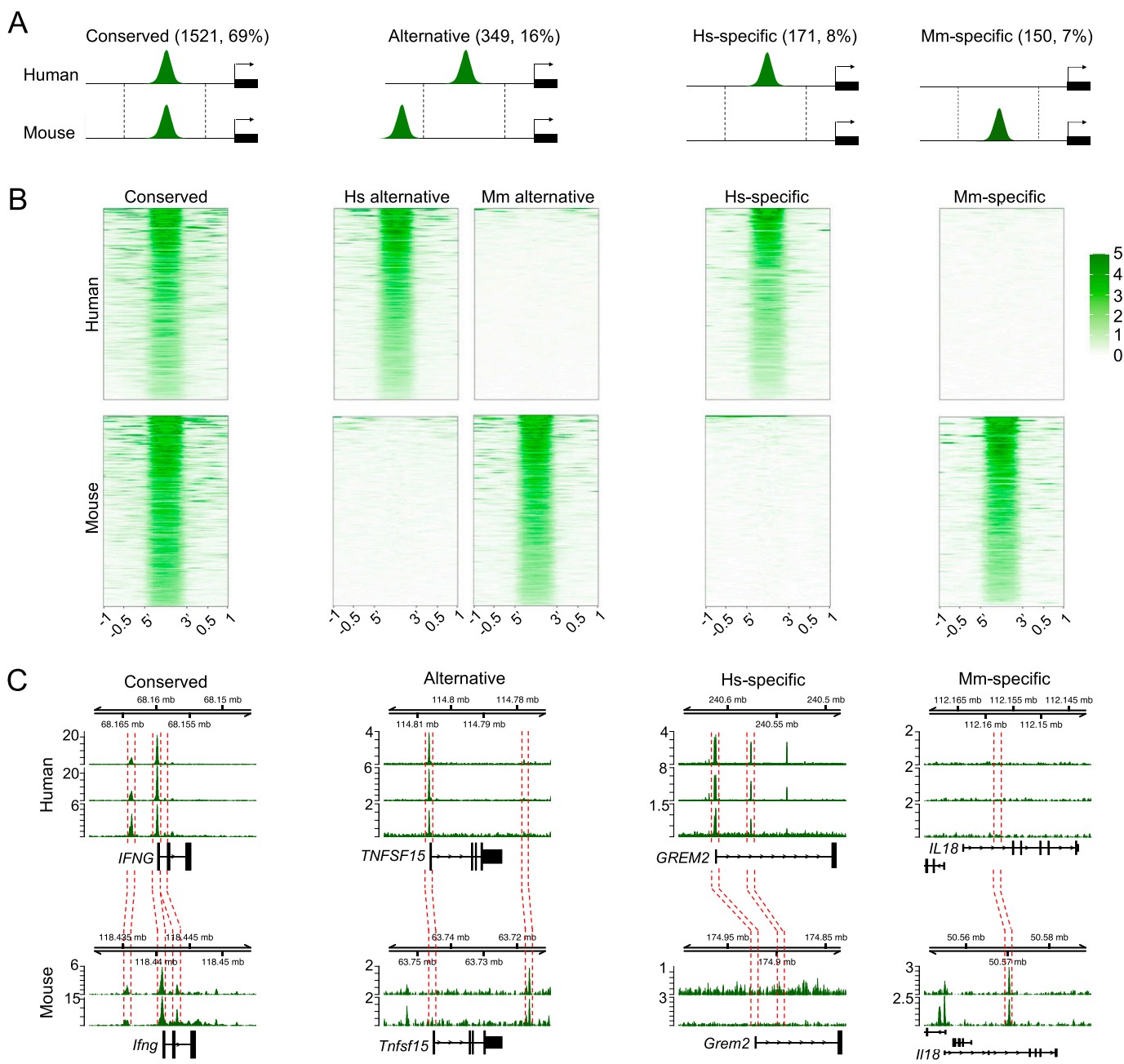

**Figure 1. Conserved and specific-specific T-bet binding in human and mouse Th1 cells.**
**(A)** Cartoon depicting four different classes of T-bet target genes and the numbers and proportions that fall into each category. Conserved target genes are defined as orthologous genes associated with one or more high-confidence T-bet binding sites at an equivalent location (defined by liftOver) in both species. Alternatively bound genes are bound by T-bet in both species but at different locations (no conserved binding sites). Hs-specific and Mm-specific target genes are only bound by T-bet in human or mouse, respectively. **(A, B)** Heat maps showing T-bet occupancy at the sets of sites described in (A). Sequence reads (per million total reads) at each position are represented by colour, according to the scale on the right (numbers of sites: Hs conserved = 2,625, Hs alternative = 611, Mm alternative liftOver to Hs = 633, Hs-specific = 222, Mm-specific liftOver to Hs = 229, Mm conserved = 2,311, Hs alternative liftOver to Mm = 648, Mm Alternative = 628, Hs-specific liftOver to Mm = 223, Mm-specific = 234). **(C)** T-bet binding at example genes with conserved, alternative, Hs-specific and Mm-specific T-bet binding. The red dashed lines show the equivalent locations of T-bet binding sites in the other species, as defined by liftOver.

$P = 4.4e^{-13}$, $t$ test versus Conserved) and, reciprocally, the genes specifically bound by T-bet in mouse tended to be more highly expressed in mouse (mean $log_2$ Hs/Mm of −1.55, $P = 0.0011$). Whereas human-specific T-bet target genes constituted 8% of T-bet target genes, they made up 53% (26 of 49) of the T-bet target genes most

highly expressed in human versus mouse ($log_2$ Hs/Mm > 5). Similarly, although mouse-specific T-bet target genes constituted 7% of T-bet target genes, they accounted for 63% (22/35) of the T-bet target genes most highly expressed in mouse versus human (both $p < 2e^{-16}$, $\chi^2$-test). Genes specifically bound by T-bet in humans and

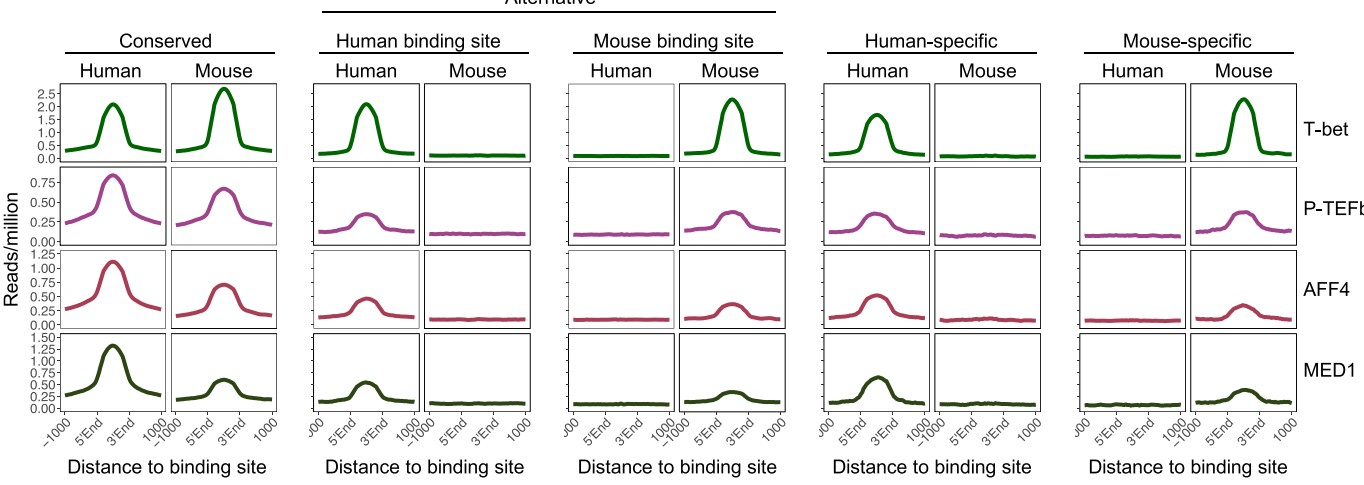

**Figure 2. Species-specific T-bet binding is associated with species-specific recruitment of P-TEFb, the super elongation complex and mediator.**
Average number of ChIP-seq reads (per million total reads) for T-bet and its co-factors P-TEFb, the super elongation complex subunit AFF4 and the Mediator subunit MED1 across conserved, alternative, human-specific and mouse-specific T-bet binding sites in human and mouse Th1 cells. Numbers of sites as in Fig 1B.

significantly (p < 1e$^{-4}$) more highly expressed in human versus mouse Th1 cells included *GREM2*, *TIMD4*, *TNFSF12*, and *PKIA* (Fig 3B). Reciprocally, genes specifically bound by T-bet in mouse and overexpressed in mouse versus human Th1 cells included *Bend4*, *Spata2*, and *Serpinb5* (Fig 3B). We conclude that species-specific T-bet occupancy is associated with species-specific gene expression.

We also examined the absolute expression level of the different classes of genes and found that genes bound specifically by T-bet in human or mouse tended to be expressed at a lower level compared with genes at which binding was conserved (p < 1 × 10$^{-14}$, *t* test; Fig S3B), which is consistent with the lower level of co-factor occupancy at specific T-bet binding sites (Fig 2).

We next considered whether there were any features of T-bet occupancy that could explain why some genes with conserved T-bet binding sites in human and mouse were nevertheless differentially expressed between the species. We found that at these genes, differences in the total number of T-bet binding sites between species were associated with differences in gene expression, with genes being more highly expressed in the species in which they were bound at the greater number of sites (linear regression of mean log$_2$ Hs/Mm expression units per T-bet binding site, p < 2e$^{-6}$; Fig 3C). This is consistent with previous observations that Th1 gene expression is driven by T-bet binding to multiple sites across extended *cis*-regulatory regions, later termed super-enhancers (12, 15, 16). Genes bound by T-bet at a greater number of sites and more highly expressed in humans than in mouse included *CASK*, *ITGAE*, and *GZMK* (Fig 3D and Table S4), whereas genes bound by T-bet at a greater number of sites and more highly expressed in mouse included *Thy1* (consistent with its known expression in mouse but not human T cells (25)), *Tex2*, and *Nfatc1* (Fig 3E and Table S4). Thus, in addition to the absolute presence and absence of T-bet binding sites, the relative number of T-bet binding sites is also associated with differential expression of Th1 genes between species.

## Species-specific T-bet binding correlates with the presence or absence of a T-bet DNA-binding motif

Like other T-box transcription factors, T-bet binds a specific DNA sequence motif (12, 13). We therefore considered whether differences in T-bet binding between human and mouse might be related to differences in the sequences at those sites between the species. To address this, we first identified consensus motifs enriched in the complete sets of high-confidence T-bet binding sites in human and mouse. This confirmed enrichment (Hs p = 1e$^{-623}$; Mm p = 1e$^{-642}$) of highly similar motifs that matched the previously determined T-bet DNA-binding motif (12) in both species (Fig 4A).

We then used FIMO (38) to quantify the proportion of conserved and species-specific T-bet binding sites that contained the T-bet DNA-binding motif. We found that the motif could be identified with confidence at roughly equal proportions of conserved T-bet binding sites in human and mouse (12.1% and 13.2%, respectively; Fig 4B). In contrast, 19.1% of human-specific T-bet binding sites contained a T-bet binding motif in human, and this dropped to 6.7% for the equivalent loci in mouse (Fig 4B). Reciprocally, 18.2% of mouse-specific T-bet binding sites contained a T-bet binding motif in mouse, and this dropped to 7.7% for the equivalent loci in human. Examination of de novo motifs enriched at species-specific binding sites revealed enrichment for a motif matching the canonical T-bet binding motif that was highly similar to the motif enriched at conserved binding sites (Fig S4A). In contrast, in the species at which T-bet was not bound at these sites, the motifs diverged from the consensus T-bet binding motif and demonstrated lower levels of enrichment. Thus, whether or not T-bet binds to a genomic location in human versus mouse correlates with whether or not the T-bet DNA binding motif is present, suggesting that differences in T-bet binding between species are due to sequence divergence at these sites.

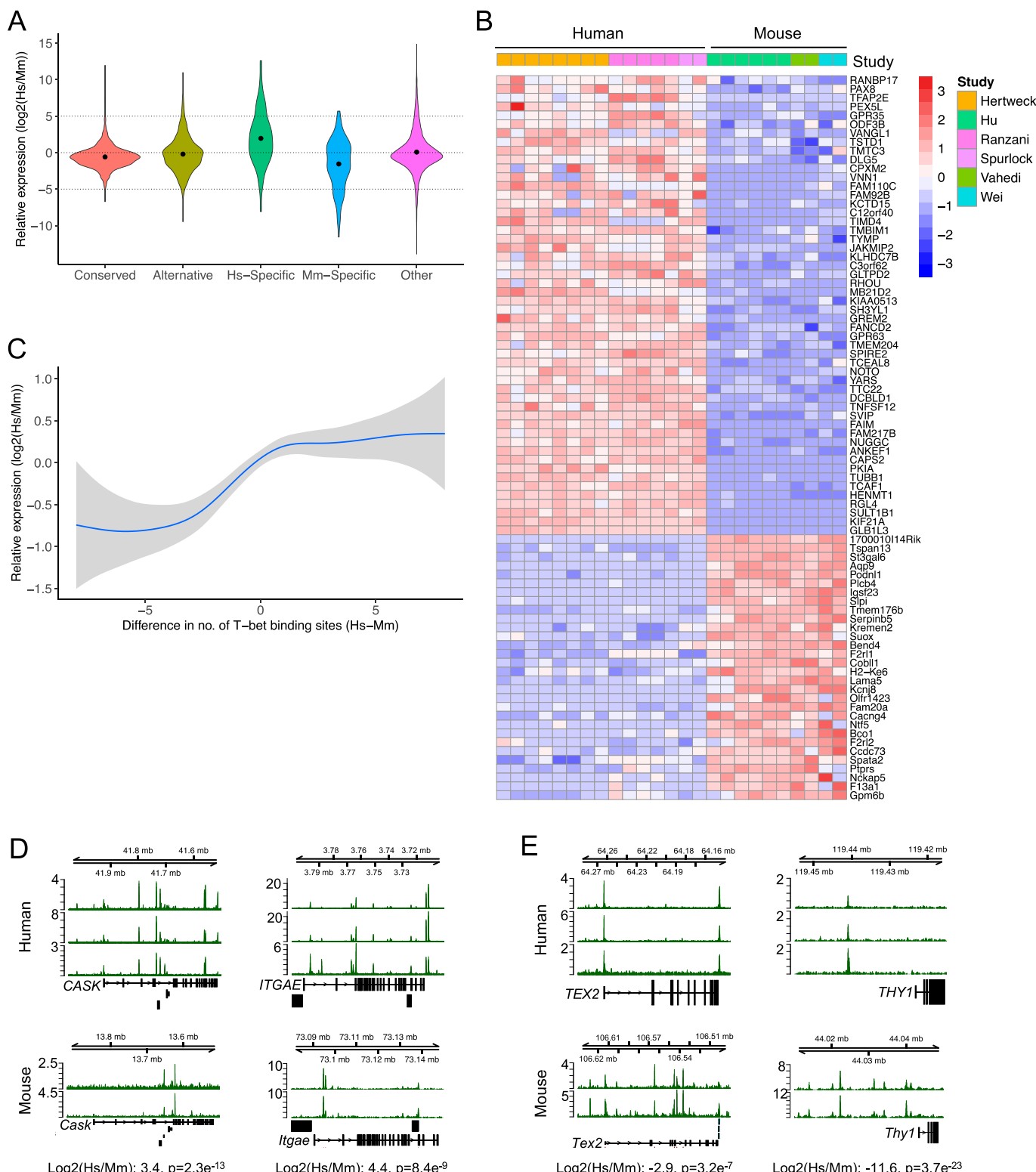

**Figure 3.   Species-specific T-bet binding is associated with species-specific Th1 gene expression.**
**(A)** Violin plot of the distribution of log₂ human versus mouse Th1 cell expression ratios for gene sets defined in Fig 1A or at other genes. Median values are marked by a dot. Mean log₂ Hs/Mm ratio for Hs-specific genes = 1.95, $P$ = 4.4e$^{-13}$ ($t$ test versus Conserved). Mean log₂ Hs/Mm for Mm-specific genes = −1.55, $P$ = 0.0011 ($t$ test versus Conserved). Numbers of genes: Conserved 1,518, Alternative 349, Hs-specific 169, Mm-specific 150, Other 13,282. **(B)** Heat map showing expression (log₂ human versus mouse expression ratio) of Hs-specific and Mm-specific genes that are significantly differentially expressed between human and mouse Th1 cells (Welch's $t$ test: unadjusted p < 1e$^{-4}$). The study from which each dataset was taken is indicated by the coloured bar at the top and the key to the right hand side. **(C)** Loess regression fit of the relation

To determine whether other transcription factor motifs were associated with species-specific T-bet binding, we compared the motifs enriched at sites bound by T-bet in both human and mouse with the motifs enriched at sites bound by T-bet in only one species or the other (Fig 4C). In the human genome, we found that human-specific sites exhibited relatively higher enrichment of RUNX motifs than conserved sites. Runx3 cooperates with T-bet to activate *Ifng* (39), suggesting that the factor may also function with T-bet at human-specific sites. In contrast, in the mouse genome, mouse-specific sites displayed relatively higher enrichment of motifs for AP-1 transcription factors, which function in T-cell activation downstream of T-cell receptor engagement (40, 41). These results suggest that, in addition to the presence or absence of T-bet binding motifs, the divergence in T-bet binding between human and mouse may reflect differences in co-factor binding.

To explore this further, we compared the motifs present at Hs-specific binding sites and Mm-specific binding sites within the same genomes. This revealed that genomic locations occupied by T-bet in mouse but not in human also exhibited enrichment for AP-1 motifs in the human genome, even though the sites were not bound by T-bet in human (Fig S4B). In contrast, T-box motifs were more highly enriched at these sites in the species in which T-bet was bound, consistent with our earlier analysis. Thus, although sites of mouse-specific T-bet binding display enrichment of AP-1 motifs, this is also apparent at these locations in the human genome and thus the presence of the motifs per se cannot be the cause of mouse-specific T-bet binding. Instead, mouse-specific T-bet binding at these sites could reflect a gain of T-box motifs in mouse and/or higher levels of AP-1 activation in mouse compared to human Th1 cells.

### Transposable elements (TEs) are enriched at species-specific T-bet binding sites

Transcription factor–binding sites can be located within TEs and TE invasions have been postulated to contribute to the evolution of regulatory gene networks (42). We therefore considered that TEs may have played a role in the diversification of T-bet binding sites between human and mouse. To test this, we compared the proportions of conserved and species-specific T-bet binding sites that overlapped TEs (Fig 5A). First looking at conserved binding sites, we found that only 3.4% overlapped a TE in humans and 0.6% in mouse. In comparison, 10.8% of human-specific and 6% of mouse-specific binding sites overlapped a TE. The enrichment of TEs at species-specific binding sites were highly significant both with a chi-square test (Hs $\chi^2$ = 142.8, Mm $\chi^2$ = 131.4, both p < $2e^{-16}$) and with permutation tests (n = 10,000, p < $1e^{-5}$) (Fig 5A). The association of species-specific binding sites with TEs was not an artefact of the genomic distribution of these sites: species-specific T-bet binding sites were enriched at distal locations compared to other T-bet binding sites

(Kruskal–Wallis test, Hs $P$ = 0.004, Mm $P$ = 0.004), but TEs did not follow the same distribution profile (Fig S5A). Breaking down TEs into their different classes revealed enrichment of LINE1 and LTR elements at species-specific binding sites compared with conserved sites (Fig 5B). Thus, these data are consistent with TE activity contributing to the divergence of T-bet binding sites between human and mouse.

To determine whether species-specific T-bet binding at TEs could be related to specific motifs within these elements, we compared the motifs enriched at T-bet binding sites that overlapped each class of TE with the motifs enriched at binding sites that did not overlap TEs (Fig S5B). In human, T-bet binding sites that overlap L1, L2, and SINE elements each exhibited enrichment of RUNX and ETS motifs. However, the enrichment of these motifs was similar to that observed at T-bet binding sites that do not overlap TEs and thus TEs are not the cause of the enrichment of RUNX motifs at Hs-specific sites. Forkhead, KLF, MafK, and some IRF/STAT motifs exhibited relatively higher enrichment at T-bet binding sites that overlap SINE elements compared to sites that do not overlap TEs but, given the small number of Hs-specific T-bet binding sites that overlap SINE elements (Fig 5B), these are unlikely to play a major role in Hs-specific T-bet occupancy. In mouse, the enrichment of AP-1 motifs was similar at T-bet binding sites that overlapped L1, LTR, and SINE elements as at binding sites that do not overlap TEs. However, other motifs did exhibit relatively higher enrichment at T-bet binding sites that overlapped specific TE classes compared to those that did not, including T-box motifs at sites that overlap L1 elements, and a subset of bZIP motifs at sites that overlap LTRs. These motifs could therefore contribute to mouse-specific T-bet binding at these TE-associated sites.

## Discussion

We have determined the degree to which the Th1 cell regulatory circuitry is conserved between human and mouse. We have found that most T-bet target genes are shared between species and that T-bet target genes associated with conserved binding sites tend to exhibit similar levels of expression. At genes with conserved binding sites, the presence of additional T-bet binding sites in human or mouse is associated with increased expression in that species. For genes at which T-bet binding sites are not conserved, it is most often the case that an alternative binding site is present at a different position in the other species and gene expression is maintained. At only a minority of genes is T-bet binding unique to human or mouse and these genes tend to be more highly expressed in the species in which T-bet is bound. Species-specific T-bet binding is associated with differences in the frequency of T-bet binding motifs. Species-specific binding sites also overlap TEs, suggesting that transposition of these elements has played a role in

between the $\log_2$ difference in human and mouse Th1 gene expression and the difference between the number of human and mouse T-bet binding sites for genes bound by T-bet in both species (grey area = 95% confidence interval). Genes with a greater number of T-bet binding sites in human tend to be more highly expressed in human, and vice versa. **(D)** Examples of genes with more T-bet binding sites and which are significantly more highly expressed in human than mouse Th1 cells. **(E)** Examples of genes with more T-bet binding sites and which are significantly more highly expressed in mouse than human Th1 cells.

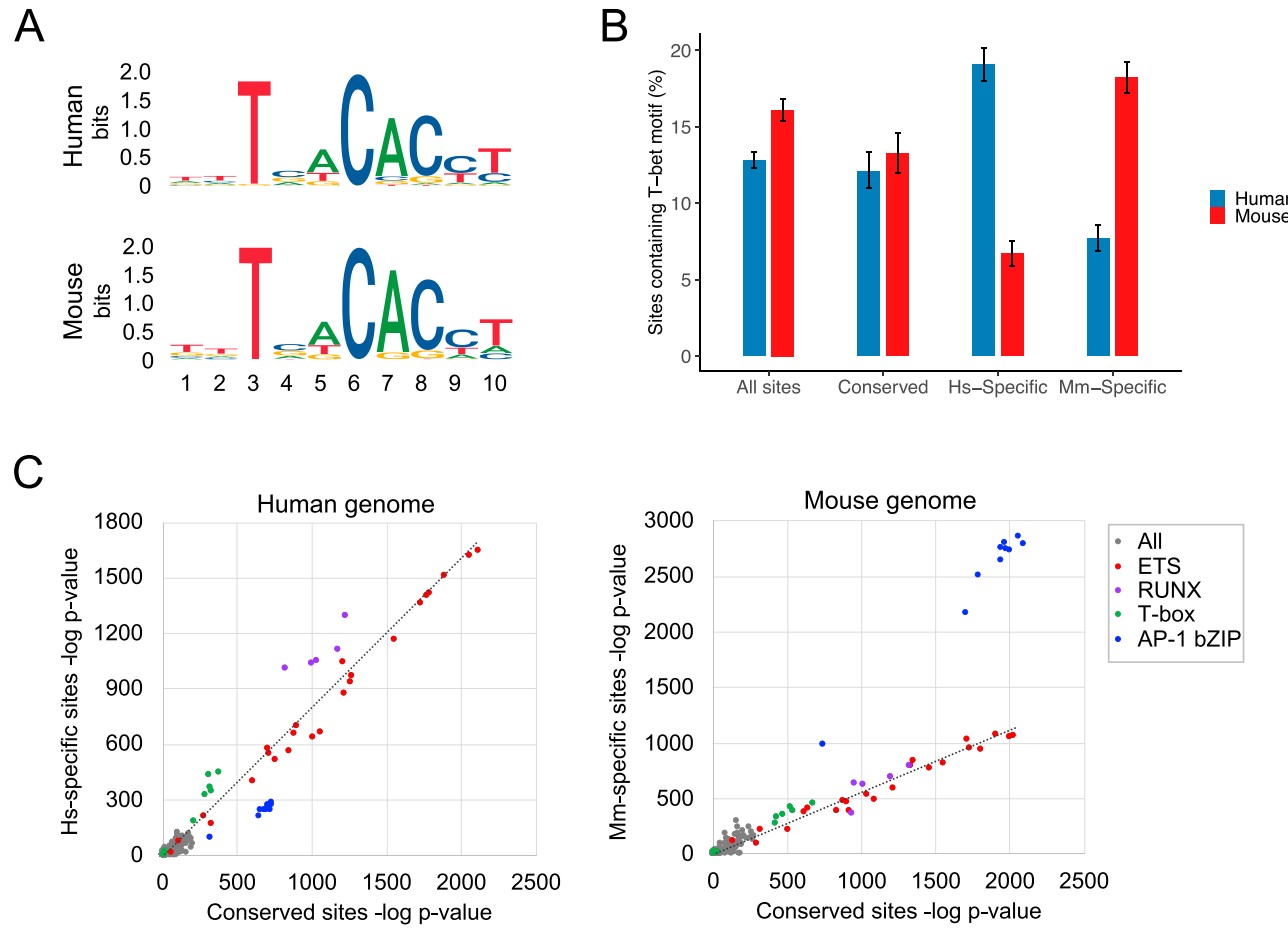

**Figure 4. Species-specific T-bet binding is associated with enrichment of DNA sequence motifs.**
**(A)** DNA binding motifs matching a previously identified consensus T-bet DNA binding motif (12) enriched in the set of T-bet binding sites in human (top) and mouse (bottom) Th1 cells. **(B)** Proportion of all T-bet binding sites, conserved T-bet binding sites, Hs-specific T-bet binding sites and Mm-specific T-bet binding sites in human and mouse that contain a sequence matching the consensus T-bet DNA binding sequence in that species (error bars = 95% confidence interval of the binomial test). **(C)** Enrichment of transcription factor binding motifs (–$\log_{10}$ P-value) within Conserved versus Hs-specific T-bet binding sites in the human genome (left) or within Conserved versus Mm-specific binding sites in the mouse genome (right). The class of the most highly enriched motifs are labelled. The dashed line shows the linear regression line for all motifs excluding AP-1 bZIP motifs.

the divergence of the Th1 cell regulatory circuitry between human and mouse.

Our analysis was designed to minimize the number of false-positive binding sites. We only considered binding sites identified at high-confidence (q < 0.01 in all replicates) and only judged a site to be species specific if the region could be identified in the other species (which was the case for 90% of sites), and there was absolutely no evidence of binding (q > 0.1 in all replicates). Application of these criteria revealed that around one-third of comparable human and mouse T-bet binding sites were conserved in the other species.

Comparison between the degree of conservation of binding sites between T-bet and those of other transcription factors is not straightforward because of differences in the criteria used to assign a position as bound or not bound between studies. However, the degree of conservation we found for T-bet is similar to that previously found for master regulator hepatocyte nuclear factor transcription factors in hepatocytes (26). Given that the immune system is subject to continuous

evolutionary pressure in the form of rapidly evolving pathogens (19, 20, 21), the similar levels of binding site conservation between T-bet and hepatocyte nuclear factor transcription factors is perhaps unexpected but reinforces the notion that the regulatory circuitry underlying the specification of T helper cell lineages is highly conserved between species.

Although one-third of T-bet binding sites are conserved between human and mouse, the proportion of T-bet target genes that are conserved is higher, with 85% of T-bet target genes bound in both species. For the vast majority of these genes, the location of at least one T-bet binding site was conserved. Furthermore, for the majority of genes at which a T-bet binding site is lost during evolution, an alternative binding site arises at the same gene. This suggests considerable pressure to conserve T-bet binding sites during human and mouse evolution. The conservation of T-bet target genes is consistent with a recent study of a human patient lacking functional T-bet protein which noted reduced numbers of immune cell types that require T-bet function in mouse, including classical Th1

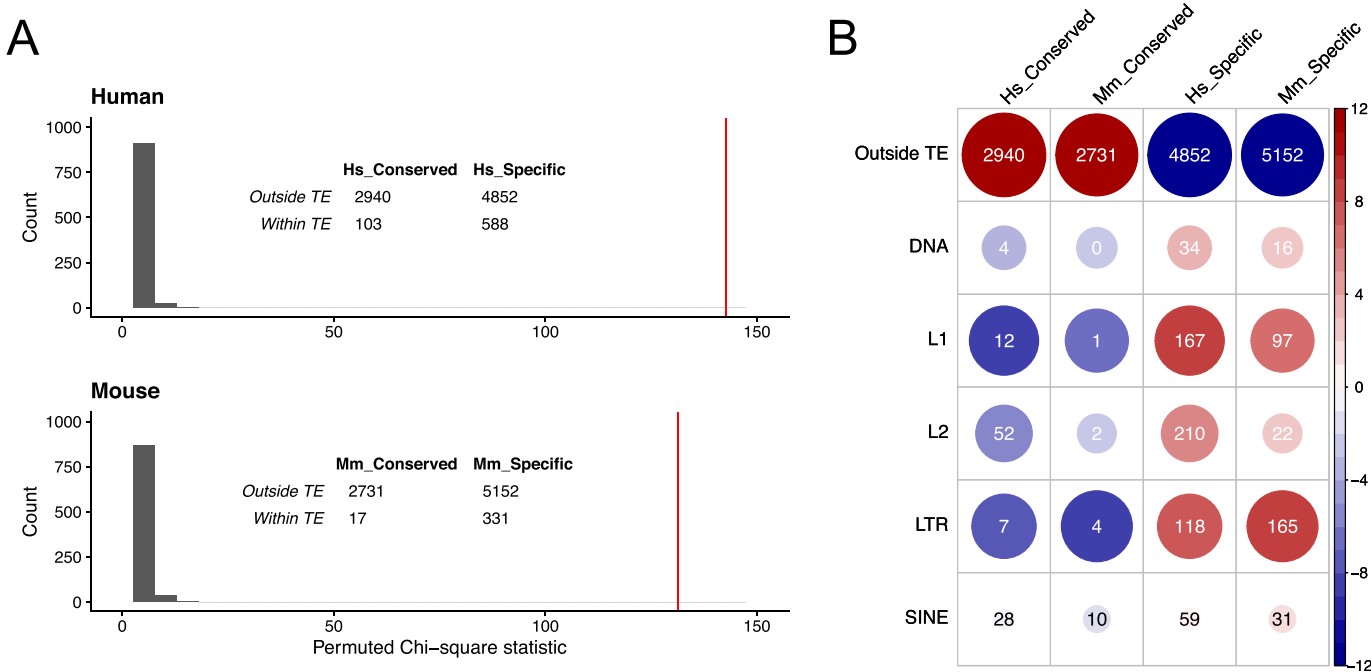

**Figure 5.  Species-specific T-bet binding sites overlap transposable elements (TEs).**
**(A)** Permutation test of the association between binding site types and TEs. In both human and mouse, species-specific binding sites are more likely to overlap a TE than conserved binding sites. The red bars show the observed overall $\chi^2$ of the inset table. The histograms show a $\chi^2$ null-distribution based on 10,000 permutations of the data. **(B)** Heat plot of the chi-square overlaps of the different classes of T-bet binding sites with different classes of TEs. The numbers show the raw table data, colour represents the standardised residuals according to the scale on the right, and circle size represents the absolute standardised residual value.

cells, NK cells, and invariant NKT cells, and reduced expression of T-bet target genes in CD4$^+$ T cells, including *IFNG*, *TNF*, *CXCR3*, *CCL3*, *CCL4*, and *XCL1* (22).

Our strategy of assigning distal T-bet binding sites to the nearest gene is likely to be imperfect and could be improved in future using promoter–enhancer interaction data from human and mouse Th1 cells. However, the relationship between species-specific T-bet binding and gene expression that we observe (Fig 3A) suggests the nearest gene model performs reasonably well.

Divergence in T-bet binding between species is correlated with divergence in co-factor recruitment and gene expression. Genes specifically bound by T-bet in human or mouse exhibit higher expression in the species in which the gene is bound; genes bound by T-bet only in humans tend to be expressed more strongly in humans and vice versa. We also found that species-specific T-bet binding sites exhibited lower occupancy of P-TEFb, AFF4, and MED1 and lower absolute levels of expression of their associated genes compared to conserved binding sites, indicating that T-bet may not have the same functionality at species-specific sites that it has at sites shared between species. Differences in the number of T-bet binding also correlates with differential expression of T-bet target genes that are shared between species, with the acquisition of additional T-bet binding sites associated with increased expression of the gene in that species. This suggests that the number of T-bet binding sites at a gene has been subjected to selective pressures and is consistent with evidence showing that transcription factor binding sites can regulate gene expression in an additive fashion (12, 15, 16, 43, 44, 45).

Species-specific T-bet binding was associated with the presence or absence of T-bet binding motifs suggesting that divergence in DNA sequence at these loci drives divergence in T-bet binding. We also identified differences in other motifs between species-specific and conserved binding sites, most strikingly for AP-1 motifs, which were enriched at mouse-specific binding sites. This enrichment of the AP-1 motif was apparent at the locations of mouse-specific T-bet binding regardless of species, suggesting that differential activation of AP-1 between human and mouse may contribute to differences in T-bet binding at these sites, rather than the presence or absence of the motif. This may be due to inherent differences in AP-1 activation between species or due to differences in the response of human and mouse Th1 cells to in vitro culture. Other differences in T-bet genome occupancy and Th1 cell biology between human and mouse may only be apparent in cells activated in response to infection in vivo.

We also found that species-specific T-bet binding sites were enriched for association with TEs, especially LINE1 and LTRs. Different classes of TEs displayed enrichment of different sets of transcription factor binding motifs, including T-box motifs and motifs for AP-1, RUNX, and ETS transcription factors, and this was more pronounced in mouse. TEs have previously been reported to be co-opted as regulatory elements in other cell types and exhibit species-specific binding of transcription factors, including STAT1/IRF1, TP53, and OCT4/NANOG (42, 46, 47, 48). Thus, in discovering enrichment of TEs at T-bet binding sites and identifying specific classes of transcription factor binding motifs at these sites, our study extends our knowledge

to include the contribution of TEs to adaptive immune cell regulatory programs.

In summary, by comparing T-bet binding and gene expression between human and mouse, we have found that the Th1 regulatory circuitry is generally conserved between species but that some key differences exist. These data will be of value in guiding the appropriate use of mice for target identification and drug development for human inflammatory, infectious and neoplastic diseases.

# Materials and Methods

## Comparison of T-bet binding data between human and mouse

Human and mouse in vitro polarised Th1 cell ChIP sequencing data were downloaded from the Gene Expression Omnibus (Hs T-bet = GSM2176976, GSM2176974, GSM776557; Mm T-bet = GSM998272, GSM836124; Hs P-TEFb = GSM1527693; Mm P-TEFb = GSM1527702; Hs AFF4 = GSM1961563; Mm AFF4 = GSM1961559; Hs MED1 = GSM1961567; Mm MED1 = GSM1961557). After trimming low-quality reads using seqtk (error rate threshold 0.05), reads were aligned to the GRCh38 or GRCm38 assemblies using Bowtie2 with default "sensitive" settings (49). High-confidence T-bet binding sites were identified by comparison to input using MACS2 (q < 0.01) (50). A high confidence set of binding sites for each species was then defined as the binding site coordinates that overlapped in all replicates. Similarly, low-confidence T-bet binding sites for each species were defined as those identified by MACS2 at q < 0.1 in any replicate. Binding sites that overlapped ENCODE blacklist regions (https://github.com/Boyle-Lab/Blacklist/tree/master/lists) were removed. The coordinates of high-confidence binding sites were extended by 1 kb either side and the equivalent coordinates identified in the other species using the mm10tohg38 and hg38tomm10 liftOver chains (from the University of California, Santa Cruz [UCSC] genome browser) and the *rtracklayer* package for R. Equivalent location was defined as a single range from the beginning to the end of the liftOver. Conserved binding sites were defined as those present at high confidence in both species and species-specific sites as those present at high confidence in one species and for which there was no evidence of binding in the other species (q > 0.1 in any replicate).

T-bet binding sites were associated with the nearest gene as defined by human GENCODE V29 or mouse GENCODE M20 transcripts annotations. Orthologous genes were identified using Ensembl Compara (51) and downloaded via Ensembl Biomart (52). Genes with conserved T-bet binding sites were defined as those associated with conserved sites in both species. Genes with alternative binding sites were defined as those associated with species-specific binding sites in both species and no conserved sites. Genes with species-specific binding were defined as those associated with a high-confidence T-bet binding site in one species and no binding sites in the other species.

## Visualisation of ChIP-seq data

We used ngsplot (53) to extract read coverage around binding sites and the equivalent regions in the other species from a single merged Binary Alignment Map file for each species and to generate average binding profiles (metagenes) and heat maps (both showing read counts per million mapped reads). To visualise T-bet binding data at individual genes, we used deeptools bamcoverage (54) to create bigwig files (read counts per million mapped reads) and then plotted these in their genomic context using the Gviz tool for R (55).

## Gene expression

RNA-seq data from in vitro polarised human and mouse Th1 cells (Table S1) were downloaded from Gene Expression Omnibus. Gene centred expression estimates were made using *kallisto* (56) with GENCODE V29 (human) and M20 (mouse) transcript models. Human and mouse expression estimates were then modelled separately using *DESeq2* (57), with experimental source and cell type treated as covariates for batch correction. Human and mouse expression data were similar in distribution but the data were zero-centered before cross-species comparison. Expression heat maps were drawn with variance stabilising transformations (*vst*). Linear regression was used to calculate the significance of the association between the difference in the number of T-bet binding sites between species (−5 and +5) and the $\log_2$ human versus mouse expression ratio.

## Motif analysis

Consensus motifs matching the previously identified T-bet DNA-binding motif (12) were identified in the complete set of high-confidence human and mouse T-bet binding sites with findMotifsGenome.pl from the HOMER tools suite (58) using the parameters hg38 or mm10—size given—mask. Conserved and species-specific T-bet binding sites were identified as before, except without extending regions by 1 kb before liftOver. The motifs were displayed with WebLogo (59). The human or mouse position-weight matrix was then used to identify significant matches ($P < 0.004$) to the T-bet consensus motif within the different sets of binding sites (all, conserved, Hs-specific and Mm-specific) for each species using FIMO (38). Confidence intervals were calculated using prop.test in R. Enriched motifs matching previously identified T-box motifs in each set of binding sites in each species were identified using findMotifsGenome.pl using the parameters hg38 or mm10—size 200—mask (use of the parameter—size given did not identify enriched motifs in the sets of liftOver coordinates). The enriched motif that matched to a T-box motif in the HOMER database was then selected. The enrichment of known motifs in each set of binding sites in each species was also calculated with findMotifsGenome.pl using the parameters hg38 or mm10—size 200—mask. The resulting $-\log_{10}$ P-values of each motif were then compared between sets of binding sites. To compare the enrichment of known motifs between sets of binding sites that overlapped classes of TE, we identified motifs significantly enriched in at least one set of binding sites (q < 0.05), normalised the $-\log_{10}$ P-values within each set of binding sites, and clustered (Euclidean distance complete linkage) and visualised the motifs with the R package pheatmap.

## TEs

Nested repeat tracks for hg38 and mm10 from Repbase (60) were downloaded from the UCSC Table Browser. Binding sites were

defined as overlapping a TE if the central 40 bp region was fully enclosed by a TE. Tests of independence were carried out using the R chisq.test function. As the numbers of Conserved and Specific sets of binding sites were different, we used a permutation test to confirm the observed $\chi^2$ value by comparing it to 10,000 permutations of the TE labels. To represent more clearly the complex associations between gene sets and particular TE types (e.g., SINE and LTR), we plotted a table of the standardised residuals of their $\chi^2$ test of independence (Observed – Expected/$\sqrt{\text{Expected}}$). Distances between binding sites or TEs and the nearest gene were taken from the midpoint of the feature to the nearest gene transcription start site.

## Supplementary Information

## Acknowledgements

This work was funded by Medical Research Council (MRC) grants to RG Jenner and GM Lord (MR/M003493/1 and MR/R001413/1), and the Cancer Research UK-University College London (CRUK-UCL) Centre (award C416/A25145). This research was also supported by the National Institute for Health (NIHR) Biomedical Research Centres at Guy's & St Thomas' NHS Foundation Trust and King's College London. The views expressed are those of the authors and not necessarily those of the NHS, the NIHR, or the Department of Health.

### Author Contributions

S Henderson: data curation, software, formal analysis, investigation, visualization, methodology, and writing—review and editing.
V Pullabhatla: data curation, formal analysis, and methodology.
A Hertweck: formal analysis, investigation, and methodology.
E de Rinaldis: supervision.
J Herrero: supervision and writing—review and editing.
GM Lord: conceptualization, funding acquisition, project administration, and writing—review and editing.
RG Jenner: conceptualization, formal analysis, supervision, funding acquisition, investigation, visualization, project administration, and writing—original draft, review, and editing.

### Conflict of Interest Statement

The authors declare that they have no conflict of interest.

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
