## [Reviewer comments · Life Science Alliance]

Life Science Alliance

The Th1 cell regulatory circuitry is largely conserved between human and mouse

Stephen Henderson, Venu Pullabhatla, Arnulf Hertweck, Emanuele de Rinaldis, Javier Herrero, Graham Lord, and Richard Jenner

DOI: <https://doi.org/10.26508/lsa.202101075>

Corresponding author(s): Richard Jenner, University College London and Graham Lord, University of Manchester

Review Timeline:

Submission Date:	2021-03-24
Editorial Decision:	2021-04-13
Revision Received:	2021-07-29
Editorial Decision:	2021-08-25
Revision Received:	2021-09-06
Accepted:	2021-09-07

Transaction Report:

April 13, 2021

Re: Life Science Alliance manuscript #LSA-2021-01075-T

Prof. Richard G Jenner
University College London
UCL Cancer Institute
Wohl Virion Centre Windeyer Institute University College London
46 Cleveland Street
London W1T 4JF

Dear Dr. Jenner,

Thank you for submitting your manuscript entitled "The Th1 cell regulatory circuitry is largely conserved between human and mouse" to Life Science Alliance. The manuscript was assessed by expert reviewers, whose comments are appended to this letter.

As you will note from the reviewers' comments below, the reviewers found your study interesting, but have also raised some important concerns, which need to be addressed prior to further consideration of the manuscript at LSA. We would, thus, like to invite you to submit a revised manuscript that addresses all of the reviewers' points.

Thank you for this interesting contribution to Life Science Alliance. We are looking forward to receiving your revised manuscript.

Sincerely,

Shachi Bhatt, Ph.D.
Executive Editor
Life Science Alliance
<http://www.lsjournal.org>
Tweet @SciBhatt @LSAJournal

- A letter addressing the reviewers' comments point by point.
- An editable version of the final text (.DOC or .DOCX) is needed for copyediting (no PDFs).
- High-resolution figure, supplementary figure and video files uploaded as individual files: See our detailed guidelines for preparing your production-ready images, <https://www.life-science-alliance.org/authors>
- Summary blurb (enter in submission system): A short text summarizing in a single sentence the study (max. 200 characters including spaces). This text is used in conjunction with the titles of papers, hence should be informative and complementary to the title and running title. It should describe the context and significance of the findings for a general readership; it should be written in the present tense and refer to the work in the third person. Author names should not be mentioned.

B. MANUSCRIPT ORGANIZATION AND FORMATTING:

Reviewer #1 (Comments to the Authors (Required)):

This manuscript by Henderson and colleagues describes a comparison of gene regulation by T-bet in human and mouse Th1 cells. T-bet is an important transcription factor that governs the differentiation of Th1 cells, which in turn are responsible for the immune response against intracellular pathogens. Understanding how the regulatory circuitry of Th1 cells is conserved along evolution is particularly important to understand how the immune system evolves under pathogen pressure, but also to address the suitability of mice as models to study the human immune response. The authors report that while only a fraction (~30%) of binding sites for T-bet are conserved

between human and mouse, in line with estimations from other transcription factors, the majority of T-bet gene targets are conserved. Expression of T-bet targets remains conserved through a combination of T-bet binding site conservation and turnover. Where T-bet binding has changed between human and mouse, gene expression tends to fall in line with the changes in transcription factor binding, with an expression increase in the species that has gained new sites. As previously described for other factors, species-specific T-bet binding sites are over-represented in transposable elements, suggesting that TEs contribute to regulatory rewiring in Th1 cells.

This study makes an interesting contribution to the growing field of interspecies comparison of gene regulatory networks, and the manuscript was particularly clear and easy to follow. I mainly have a question regarding the fraction of sites that could actually be tested (vs. defined as indeterminate due to either poor sequence conservation across species or intermediate q-value calls), and a number of more minor comments, but altogether the manuscript seems suitable for publication in LSA.

1. The first result section should include quantitative information regarding how many T-bet ChIP-seq replicates were available in each species, how many peaks were called in each species (reproducibly and non-reproducibly), how many of those could be transferred between reference genomes by Lifter, and how many of these were called as indeterminate because of an intermediate q-value in one of the species. This information is partly assessable by browsing the supplementary tables, so I am fairly confident that the analysis is robust, but these numbers are important for transparency and to interpret the effect sizes, p-values, etc, along the analysis.
2. Assignment of T-bet binding sites to gene targets was done by a nearest-gene approach, which has been shown to be quite unreliable except for TFs that bind close to the TSS. This should be acknowledged more explicitly in the main text, or perhaps backed up with a supplementary figure if most T-bet bind in a gene-proximal manner. Additionally, what was the distribution of site counts per target? This could also further reinforce that most targets are well-identified, if many genes are predicted as targets of multiple binding sites.
3. Fig 2 shows that there are fairly sizeable differences in read coverage across different categories of binding sites. Are those y-axes comparable in any way, and if so, do some categories of binding sites exhibit higher coverage on average, suggestive of more systematic/stronger binding?
4. Fig 3A: is there any difference in expression levels between the targets of conserved/alternative/specific T-bet binding sites? Could this analysis be confounded by differences in average gene expression between the different categories of targets?
5. The number of T-bet binding sites carrying the canonical motif seems low (<15%). Is this in line with previous reports in the literature?
6. Fig 4C and D: it is unclear to me how the permutation test was performed (about 50% of human and mouse genomes are TEs, so I'd expect TEs to overlap randomized T-bet sites more, rather than less, frequently than observed). The numbers in plot D are not interpretable, they are too small to be meaningful (e.g. I am unconvinced that 9 Mm-alternate TFBS in SINEs represent an enrichment). Please either scrap or strongly temper the corresponding text (p10, l.1-5).
7. I don't think it is entirely fair to conclude on p.10 L.10 that "the majority" of T-bet binding sites are conserved between species as I expect that the majority of sites in either species were not included in the analysis, because they cannot be mapped between species. This needs to be

discussed, as well as the fact that binding sites in TEs are likely to be disproportionately represented in this non-considered set, as TE-rich regions are notably more difficult to align between species.

Reviewer #2 (Comments to the Authors (Required)):

The authors report an analysis of genomic binding locations (and putative target genes) for the lineage-determining transcription factor Tbx21 in human and mouse cells. Specifically, the manuscript leverages publicly available ChIP- and RNA-sequencing datasets in human and mouse Th1 cells to compare genomic binding locations of Tbx21, and investigate the properties of binding locations common to the two species or specific to one of the two. The manuscript reports a high correspondence of Tbx21 binding locations between human and mouse cells, although binding sites specific to either human or mouse also recruit T-bet dependent co-factors. By associating Tbx21 binding regions to proximal gene expression, the authors report an association between gene expression and Tbx21 binding both for conserved sites (associated to similar expression in both species) and sites specific to one species (associating with differential gene expression). Lastly, the study investigates what properties of Tbx21 binding may explain these observations, including the relative number of binding sites, the presence of consensus Tbx21 binding motifs and the overlap between Tbx21 binding locations and annotated transposable elements.

Overall, this is an interesting analysis making use of publicly available datasets to explore a question relevant to gene regulation in Th1 cells and its divergence between human and mouse. However, the analyses reported in some sections of the paper are somewhat superficial, and could be substantially revised to strengthen the conclusions of the manuscript. Below, we detail major and minor points we feel the authors should consider in revising their manuscript.

Major points:

1) The manuscript text and figures should be revised to improve the reporting of analyses, in particular with regards to experimental design and absolute numbers of regions in various categories and plots. Specifically, Figure 1 should include a panel detailing very clearly what the Tbx21 ChIP-seq dataset actually is, including details on:
(i) how Th1 cells have been extracted in each species (e.g. are these ex vivo experiments, or have the cells being cultured/derived in vitro from ex-vivo populations? For gene expression datasets, are the cells from the six studies of a similar origin/treatment?), (ii) number of replicates in each species' ChIP-seq datasets, and (iii) total number of peaks/binding locations used for analysis in each species.

The manuscript often refers to percentages, but absolute numbers should also be included in the main text and figures for clarity. Some examples:
- panels A and B of Figure 1 (how many binding locations are in each of the four categories in A, and in each of the 10 heatmaps in B?).
- main text, such as lines 3-4 in page 5 (how many sites were considered as indeterminate and not considered for further analyses in each species?).
- panel A in Figure 3 (how many sites are represented in each category/violin plot?)

2) Some of the analyses appear somewhat superficial, and as a result the conclusions in several of the results sections could be strengthened with revised analyses (in particular, sections 2, 4 and 5). First, the analysis in section 2 focuses on the co-occupancy of Tbx21 binding locations by T-bet dependent co-factors. Albeit interesting, the conclusion that binding of Tbx21 exclusively in human

or mouse also results in recruitment of co-activators is somewhat expected. These analyses (see also comments below) could be strengthened by more comprehensive evaluation of what properties may distinguish conserved and species-specific Tbx21 binding regions, including at the sequence level. Following on the results reported in this section, the authors could investigate whether the various categories in Figure 1A have any differences in enriched transcription factor binding motifs, which may for instance be indicative of transcription factor cooperativity (see Stefflova et al. Cell 2013 for a related analysis). The authors could also consider integrating this section with section 4 (focusing on Tbx21 binding motifs).

Second, analyses in section 4 focus on the presence or absence of Tbx21 binding motifs in conserved or species-specific sites. The reported results are somewhat counterintuitive in that binding regions exclusive to either human or mouse show a higher proportion of Tbx21 binding motifs. However, this analysis appears to be based on searches for the Tbx21 consensus binding sequence (i.e. with FIMO). An alternative interpretation of these results is that conserved binding regions may favour a slightly altered Tbx21 consensus sequence, for instance as a result of cooperative interactions with other transcription factors. This could be evaluated with de novo motif finding tools such as MEME, and integrated with analyses suggested above.

Lastly, a more careful consideration of the sequence-level properties of Tbx21 binding regions could also allow the authors to expand their results in Figure 4 to consider whether any of the observed enrichments of transposable elements (TEs) associates with specific transcription factor binding sequences. Similarly, the gene expression levels of genes proximal to Tbx21 binding regions overlapping TEs could have been analysed - are this subset of sites significantly different from the non-TE fraction in their association to proximal gene expression?

Minor points:

- The authors may want to avoid the "species-specific" terminology in the manuscript, as most genomic regions bound only in either mouse or human cells (in a two-species comparison) will NOT be species-specific sequences. A more agnostic terminology for these regions could be "human-only" or "mouse-only" instead.
- The representation of differentially expressed genes in 3B could be improved. As the authors are effectively integrating six gene expression datasets, they could consider substituting or combining this panel with a summary view of differentially expressed genes (based on the integration of three datasets in each species).
- The analyses largely ignore whether properties of Tbx21 binding regions are independent of their distance to the TSS (or not). This could be evaluated in several sections and may strengthen some of the conclusions. One example is in lines 17-18 in page 7 - is the association of species-specific T-bet occupancy with differential gene expression more pronounced for sites proximal to a TSS, or independent of this?
- The discussion section should carefully consider likely limitations of the study. If we understand correctly, the ChIP-sequencing datasets analysed here correspond to Th1 cells in their steady-state, rather than in response to specific pathogens or infections. Conceptually, the latter could modify the Tbx21 binding repertoire, and these stimulus-specific binding regions may have modified patterns of divergence between human and mouse. The authors should consider including a related discussion in the manuscript, especially if there are relevant datasets/previous observations in the field.
- The referencing of the manuscript is somewhat sparse in some sections, and the authors should consider some revision of the references (e.g. line 12 in page 5, reference could be included for "classical Th1 genes")
- It would be helpful to specify accession numbers of RNA-sequencing datasets in the methods, similarly as is done for the ChIP-sequencing experiments (lines 16-17, page 14).

The Th1 cell regulatory circuitry is largely conserved between human and mouse**Response to reviewers****Reviewer #1**

Reviewer: This manuscript by Henderson and colleagues describes a comparison of gene regulation by T-bet in human and mouse Th1 cells. T-bet is an important transcription factor that governs the differentiation of Th1 cells, which in turn are responsible for the immune response against intracellular pathogens. Understanding how the regulatory circuitry of Th1 cells is conserved along evolution is particularly important to understand how the immune system evolves under pathogen pressure, but also to address the suitability of mice as models to study the human immune response.

The authors report that while only a fraction (~30%) of binding sites for T-bet are conserved between human and mouse, in line with estimations from other transcription factors, the majority of T-bet gene targets are conserved. Expression of T-bet targets remains conserved through a combination of T-bet binding site conservation and turnover. Where T-bet binding has changed between human and mouse, gene expression tends to fall in line with the changes in transcription factor binding, with an expression increase in the species that has gained new sites. As previously described for other factors, species-specific T-bet binding sites are over-represented in transposable elements, suggesting that TEs contribute to regulatory rewiring in Th1 cells.

This study makes an interesting contribution to the growing field of interspecies comparison of gene regulatory networks, and the manuscript was particularly clear and easy to follow. I mainly have a question regarding the fraction of sites that could actually be tested (vs. defined as indeterminate due to either poor sequence conservation across species or intermediate q-value calls), and a number of more minor comments, but altogether the manuscript seems suitable for publication in LSA.

Response: We thank the reviewer for their careful review of our manuscript and are pleased that they found the paper to make an interesting contribution to the field.

1. The first result section should include quantitative information regarding how many T-bet ChIP-seq replicates were available in each species, how many peaks were called in each species (reproducibly and non-reproducibly), how many of those could be transferred between reference genomes by Lifter, and how many of these were called as indeterminate because of an intermediate q-value in one of the species. This information is partly assessable by browsing the supplementary tables, so I am fairly confident that the analysis is robust, but these numbers are important for transparency and to interpret the effect sizes, p-values, etc, along the analysis.

Response: We apologise that this information was not easily accessible. We have now added these numbers to new Figure S1A, which displays them as a flow-chart for clarity in how the different sets of binding sites are related to each other.

2. Assignment of T-bet binding sites to gene targets was done by a nearest-gene approach, which has been shown to be quite unreliable except for TFs that bind close to the TSS. This should be acknowledged more explicitly in the main text, or perhaps backed up with a supplementary figure if most T-bet bind in a gene-proximal manner. Additionally, what was the distribution of site counts per target? This could also further reinforce that most targets are well-identified, if many genes are predicted as targets of multiple binding sites.

The Th1 cell regulatory circuitry is largely conserved between human and mouse

Response: We agree that the nearest-gene approach is not perfect but is the only option in the absence of promoter capture HiC data for Th1 cells. We have now added discussion of the limitation of this approach to the Discussion (p.14 line 10). However, even with the uncertainty inherent in the nearest-gene method, we still observe a significant relationship between species-specific T-bet binding and gene expression (Figure 3A). We also demonstrated that genes with a greater number of binding sites in one species exhibit higher expression in that species (Figure 3C), suggesting reasonably accurate target gene identification overall. We thank the reviewer for the suggestion of calculating the distribution of T-bet binding site counts per target gene. This shows that 45% of T-bet target genes are associated with more than one T-bet binding site in human and mouse (Figure 1, below).

Figure 1. Distribution of number of T-bet binding sites (peaks) per gene in Hs (top) and Mm (bottom) identified using the nearest-gene approach.

3. Fig 2 shows that there are fairly sizeable differences in read coverage across different categories of binding sites. Are those y-axes comparable in any way, and if so, do some categories of binding sites exhibit higher coverage on average, suggestive of more systematic/stronger binding?

Response: For each co-factor, the y-axes are comparable between the different categories of binding sites within a species. The reviewer is correct that the different categories of binding sites exhibit different levels of co-factor occupancy. Although T-bet occupancy was similar at alternative and species-specific sites compared to conserved sites, P-TEFb, AFF4 and MED1 occupancy was reduced at alternative and species-specific sites (Figures 2 and S2), suggesting that T-bet may not exhibit the same functionality at these sites as it does at conserved sites. We have now added this observation to the Results (p.6 line 19) and Discussion (p.14 line 19).

4. Fig 3A: is there any difference in expression levels between the targets of conserved/alternative/specific T-bet binding sites? Could this analysis be confounded by differences in average gene expression between the different categories of targets?

The Th1 cell regulatory circuitry is largely conserved between human and mouse

Response: Genes associated with conserved or alternative binding sites have higher absolute levels of expression than genes only associated with Hs-specific or Mm-specific binding sites (new Figure S3B and p.8 line 6). This is consistent with the lower levels of co-factor occupancy at species-specific T-bet binding sites compared with conserved binding sites (point 3 above). However, we don't believe this confounds the interpretation of differential gene expression between species; although Hs-specific and Mm-specific genes have similar distributions of absolute expression, they exhibit completely opposite patterns of differential expression between species (Figure 3A).

5. *The number of T-bet binding sites carrying the canonical motif seems low (<15%). Is this in line with previous reports in the literature?*

Response: Figure 4 shows the proportion of sites that contain a highly significant match ($p < 0.004$) to the human or mouse motif as identified by FIMO. The proportion of sites that contain the motif could therefore well be higher. A similar proportion of T-bet binding sites were previously found to contain the canonical motif by Kanhere et al., 2012 (Nat Commun 3:1268). The motifs being searched against were identified from the complete sets of high-confidence human and mouse T-bet binding sites by HOMER and both significantly enriched in these sets of sites (human $p = 1e-623$, mouse $p = 1e-642$, p.9 line 11).

6. *Fig 4C and D: it is unclear to me how the permutation test was performed (about 50% of human and mouse genomes are TEs, so I'd expect TEs to overlap randomized T-bet sites more, rather than less, frequently than observed). The numbers in plot D are not interpretable, they are too small to be meaningful (e.g. I am unconvinced that 9 Mm-alternate TFBS in SINEs represent an enrichment). Please either scrap or strongly temper the corresponding text (p10, l.1-5).*

Response: Whilst a high proportion of the genome is made up of TEs, we don't expect functional elements such as T-bet binding sites to be randomly distributed. Thus, rather than randomising the location of T-bet binding sites, we compared the overlap between TEs and conserved T-bet binding sites with the overlap between TEs and species-specific T-bet binding sites and confirmed significance with a permutation test. The chi-square p-value for the overall table was $2e-14$ and for SINE elements the p-value was 0.00077.

However, although significant, we acknowledge that the number of sites in this analysis was low (we only included sites at genes that could be categorised as shown in Figure 1A). We have now repeated the analysis with all conserved and species-specific T-bet binding sites as this provides larger sample sizes and is more consistent with the analyses in Figure 4. This new analysis (now Figure 5) confirms that species-specific binding sites are more likely than conserved sites to overlap TEs, especially LINE1 and LTR elements. We have modified the text (p.11, lines 6-21) to accompany this new figure.

7. *I don't think it is entirely fair to conclude on p.10 L.10 that "the majority" of T-bet binding sites are conserved between species as I expect that the majority of sites in either species were not included in the analysis, because they cannot be mapped between species. This needs to be discussed, as well as the fact that binding sites in TEs are likely to be disproportionately represented in this non-considered set, as TE-rich regions are notably more difficult to align between species.*

Response: We thank the reviewer for highlighting the need for us to correct this statement. This should have echoed the abstract by stating that the majority of T-bet target *genes* are conserved between species. As stated on p.5 line 6, and also now shown in Figure S1A,

The Th1 cell regulatory circuitry is largely conserved between human and mouse

around 1/3 of T-bet binding sites are conserved in the other species. This number does not change significantly when taking into account sites that cannot be mapped between species as these only account for 10% of binding sites (Figure S1A), which we have now clarified in the results (p.5 line 1) and Discussion (p.13 line 10). As described on p.5 line 10, to compare T-bet gene targeting between species, we focused on only those binding sites associated with orthologous genes. This identified 2912 genes bound by T-bet in human or mouse of which the majority (69%) were associated with a conserved T-bet binding site.

Reviewer #2

Reviewer: The authors report an analysis of genomic binding locations (and putative target genes) for the lineage-determining transcription factor Tbx21 in human and mouse cells. Specifically, the manuscript leverages publicly available ChIP- and RNA-sequencing datasets in human and mouse Th1 cells to compare genomic binding locations of Tbx21, and investigate the properties of binding locations common to the two species or specific to one of the two. The manuscript reports a high correspondence of Tbx21 binding locations between human and mouse cells, although binding sites specific to either human or mouse also recruit T-bet dependent co-factors. By associating Tbx21 binding regions to proximal gene expression, the authors report an association between gene expression and Tbx21 binding both for conserved sites (associated to similar expression in both species) and sites specific to one species (associating with differential gene expression). Lastly, the study investigates what properties of Tbx21 binding may explain these observations, including the relative number of binding sites, the presence of consensus Tbx21 binding motifs and the overlap between Tbx21 binding locations and annotated transposable elements.

Overall, this is an interesting analysis making use of publicly available datasets to explore a question relevant to gene regulation in Th1 cells and its divergence between human and mouse. However, the analyses reported in some sections of the paper are somewhat superficial and could be substantially revised to strengthen the conclusions of the manuscript. Below, we detail major and minor points we feel the authors should consider in revising their manuscript.

Response: We are pleased that the reviewer found our work to present an interesting analysis and thank them for their advice in how to strengthen the manuscript.

Major points:

Reviewer: 1) *The manuscript text and figures should be revised to improve the reporting of analyses, in particular with regards to experimental design and absolute numbers of regions in various categories and plots. Specifically, Figure 1 should include a panel detailing very clearly what the Tbx21 ChIP-seq dataset actually is, including details on: (i) how Th1 cells have been extracted in each species (e.g. are these ex vivo experiments, or have the cells being cultured/derived in vitro from ex-vivo populations? For gene expression datasets, are the cells from the six studies of a similar origin/treatment?), (ii) number of replicates in each species' ChIP-seq datasets, and (iii) total number of peaks/binding locations used for analysis in each species.*

The manuscript often refers to percentages, but absolute numbers should also be included in the main text and figures for clarity. Some examples:

- panels A and B of Figure 1 (how many binding locations are in each of the four categories in A, and in each of the 10 heatmaps in B?).

- main text, such as lines 3-4 in page 5 (how many sites were considered as indeterminate and not considered for further analyses in each species?).

- panel A in Figure 3 (how many sites are represented in each category/violin plot?)

The Th1 cell regulatory circuitry is largely conserved between human and mouse

Response: The identities of the ChIP-seq and RNA-seq datasets were provided in the Methods but we apologise for not describing these in the manuscript. All of the ChIP-seq and RNA-seq datasets are Th1 cells polarised (differentiated) *in vitro* from purified naïve or total CD4⁺ T cells. We have now included details on the methods used to generate each sample in new Table S1. Importantly, except for the purification of T cells from blood in humans and lymph nodes and spleen in mouse, there are no consistent differences between how the mouse and human Th1 cells were prepared for either ChIP-seq or RNA-seq.

The absolute numbers for the categories in Figures 1A and B were given in the results (Conserved 1521 genes, Alternative 349 genes, Hs-specific 171 genes, Mm-specific 150 genes) but these numbers have now been added to Figure 1A for clarity. The heatmaps in Figures 1B and S2 and metagene profiles in Figure 2 show protein occupancy at the T-bet binding sites at these genes and the numbers of sites have now been added to the legends. The numbers of indeterminant binding sites in human and mouse, together with the number of other types of binding site, has been clarified in new Figure S1A. The numbers of genes in each category in Figure 3A has now been added to the legend. The number of T-bet binding sites overlapping each class of TE is stated in Figure 5B. The number of genes present in Figure S3A has now been added to the legend.

Reviewer: 2) *Some of the analyses appear somewhat superficial, and as a result the conclusions in several of the results sections could be strengthened with revised analyses (in particular, sections 2, 4 and 5).*

First, the analysis in section 2 focuses on the co-occupancy of Tbx21 binding locations by T-bet dependent co-factors. Albeit interesting, the conclusion that binding of Tbx21 exclusively in human or mouse also results in recruitment of co-activators is somewhat expected. These analyses (see also comments below) could be strengthened by more comprehensive evaluation of what properties may distinguish conserved and species-specific Tbx21 binding regions, including at the sequence level. Following on the results reported in this section, the authors could investigate whether the various categories in Figure 1A have any differences in enriched transcription factor binding motifs, which may for instance be indicative of transcription factor cooperativity (see Stefflova et al. Cell 2013 for a related analysis). The authors could also consider integrating this section with section 4 (focusing on Tbx21 binding motifs).

Response: We respectfully disagree that the results of Figure 2 are expected and believe that these data demonstrate the important point that there is no redundancy. The loss of T-bet binding at a site in human or mouse is accompanied by a loss of activity of the associated regulatory element and therefore demonstrates that other factors are not able to compensate for the loss of T-bet function at these sites. We have added new text to the results (p.6 line 17) to clarify this point. We thank the reviewer for suggesting we perform an analysis of the motifs present at these different types of T-bet binding sites and we have now undertaken such a comparison (new Figures 4C and S4B, p.10 line 5). We found that human-specific sites exhibited relatively higher enrichment of RUNX motifs compared with conserved sites. Runx3 cooperates with T-bet to activate *Ifng* (Djuretic et al., 2007. *Nat Immunol.* 8: 145-53), suggesting that the factor may also function with T-bet at human-specific sites. In contrast, mouse-specific sites displayed relatively higher enrichment of motifs for AP-1 transcription factors, which function in T cell activation downstream of T cell receptor engagement. Interestingly, although sites of mouse-specific T-bet binding display enrichment of AP-1 motifs, this is also apparent at these locations in the human genome and thus the presence of the motifs *per se* cannot be the cause of mouse-specific T-bet binding. Instead, mouse-specific T-bet binding at these sites could instead reflect a gain of T-box motifs in mouse and/or higher levels of AP-1 activation in mouse compared to human Th1 cells.

The Th1 cell regulatory circuitry is largely conserved between human and mouse

Reviewer: *Second, analyses in section 4 focus on the presence or absence of Tbx21 binding motifs in conserved or species-specific sites. The reported results are somewhat counterintuitive in that binding regions exclusive to either human or mouse show a higher proportion of Tbx21 binding motifs. However, this analysis appears to be based on searches for the Tbx21 consensus binding sequence (i.e. with FIMO). An alternative interpretation of these results is that conserved binding regions may favour a slightly altered Tbx21 consensus sequence, for instance as a result of cooperative interactions with other transcription factors. This could be evaluated with de novo motif finding tools such as MEME, and integrated with analyses suggested above.*

Response: That a higher proportion of species-specific sites contain the consensus T-bet binding motifs is perhaps not too surprising when one considers that the consensus motifs were derived from the complete set of high-confidence binding sites in each species and that there are a greater number of species-specific sites than conserved sites (see new Figure S1A), and thus will have a greater influence on the consensus. We thank the reviewer for the suggestion of comparing the T-bet consensus sequence between conserved and species-specific binding sites and we have now performed this analysis (new Figure S4A and p.9 line 20). We found that in the species in which T-bet was present, the motif enriched at species-specific binding sites was highly similar to that enriched at conserved binding sites. In contrast, in the species at which T-bet was not bound at these sites, the enriched motif that most closely matched a T-box binding element diverged from the consensus T-bet binding motif. Thus, these new results are consistent with our previous analysis and together demonstrate that species-specific T-bet binding is associated with differences in the proportion of sequences that contain T-bet binding motifs between species.

Reviewer: *Lastly, a more careful consideration of the sequence-level properties of Tbx21 binding regions could also allow the authors to expand their results in Figure 4 to consider whether any of the observed enrichments of transposable elements (TEs) associates with specific transcription factor binding sequences. Similarly, the gene expression levels of genes proximal to Tbx21 binding regions overlapping TEs could have been analysed - are this subset of sites significantly different from the non-TE fraction in their association to proximal gene expression?*

Response: We thank the reviewer for the suggestion of determining whether binding sites that overlap TEs exhibit differences in motifs or differences in expression of the associated genes. We have now performed these analyses. We did not observe any differences in the expression of genes associated with different classes of T-bet binding sites divided by TE association (Figure 2, below). We did observe some differences in the patterns of motif enrichment between T-bet binding sites that overlapped TEs versus sites that did not (new Figure S5B and p.11 line 23). In human, T-bet binding sites that overlap L1, L2 and SINE elements each exhibited enrichment of RUNX and ETS motifs. However, the enrichment of these motifs was similar to that observed at T-bet binding sites that do not overlap TEs and thus TEs are not the cause of the enrichment of RUNX motifs at Hs-specific sites. Forkhead, KLF, MafK and some IRF/STAT motifs exhibited relatively higher enrichment at T-bet binding sites overlapping SINE elements compared to sites that do not overlap TEs but, given the small number of Hs-specific T-bet binding sites that overlap SINE elements, these are unlikely to play a major role in Hs-specific T-bet occupancy. In mouse, the enrichment of AP-1 motifs was similar at T-bet binding sites that overlapped L1, LTR and SINE elements as at binding sites that don't overlap TEs. Other motifs exhibited relatively higher enrichment at murine T-bet binding sites that overlapped specific TE classes compared to those that did not, including T-box motifs at sites that overlap L1 elements and a subset of bZIP motifs at sites that overlap LTRs. These motifs could therefore contribute to mouse-specific T-bet binding at these sites.

The Th1 cell regulatory circuitry is largely conserved between human and mouse

Figure 2. Violin plot of the distribution of log₂ human vs mouse Th1 cell expression ratios for gene sets defined in Figure 1A, split into whether or not the T-bet binding site overlaps a TE in human (left) or mouse (right). Median values are marked by a dot.

Minor points:

Reviewer: *The authors may want to avoid the "species-specific" terminology in the manuscript, as most genomic regions bound only in either mouse or human cells (in a two-species comparison) will NOT be species-specific sequences. A more agnostic terminology for these regions could be "human-only" or "mouse-only" instead.*

Response: We referred to species-specific binding rather than species-specific sequences and we have now further clarified this in the text (p.5, line 2)

Reviewer: *The representation of differentially expressed genes in 3B could be improved. As the authors are effectively integrating six gene expression datasets, they could consider substituting or combining this panel with a summary view of differentially expressed genes (based on the integration of three datasets in each species).*

Response: We have now marked the position of genes with species-specific binding in Figure S3A, which shows relative levels of expression of all genes.

Reviewer: *The analyses largely ignore whether properties of Tbx21 binding regions are independent of their distance to the TSS (or not). This could be evaluated in several sections and may strengthen some of the conclusions. One example is in lines 17-18 in page 7 - is the association of species-specific T-bet occupancy with differential gene expression more pronounced for sites proximal to a TSS, or independent of this?*

Response: We have previously investigated the relationship between the location of T-bet binding sites and gene expression and found that T-bet transactivation function is most pronounced at genes with multiple distal binding sites (Kanhere et al., 2012. Nat Commun 3:1268). Figure 3C of this new study builds upon this observation by showing that genes bound by T-bet in both human and mouse are more highly expressed in the species that has the greater number of binding sites.

Reviewer: *The discussion section should carefully consider likely limitations of the study. If we understand correctly, the ChIP-sequencing datasets analysed here correspond to Th1 cells in their steady-state, rather than in response to specific pathogens or infections.*

The Th1 cell regulatory circuitry is largely conserved between human and mouse

Conceptually, the latter could modify the Tbx21 binding repertoire, and these stimulus-specific binding regions may have modified patterns of divergence between human and mouse. The authors should consider including a related discussion in the manuscript, especially if there are relevant datasets/previous observations in the field.

Response: We thank the reviewer for prompting discussion of this important point. For all of the ChIP-seq datasets, and most of the RNA-seq datasets used in the study, the Th1 cells were generated by activation of CD4⁺ T cells with anti-CD3/CD28 in Th1 polarising conditions (IL12 and anti-IL4), which are intended to mimic the conditions that induce differentiation of these cells *in vivo*. There is no evidence from the literature that gene expression differs between Th1 cells activated *in vitro* and Th1 cells activated in response to infection *in vivo*. For five of the human RNA-seq datasets, the Th1 cells were instead polarised *in vivo* and purified by means of the marker CXCR3 (new Table S1 and Ranzani et al., 2015) and these display a similar expression profile to the *in vitro* polarised samples (compare Ranzani to other human samples in Figure S3A). However, we agree that it is possible that T-bet occupies different sets of sites during the acute response to infection *in vivo* and we have now acknowledged this limitation in the Discussion (p.15 line 13).

- The referencing of the manuscript is somewhat sparse in some sections, and the authors should consider some revision of the references (e.g. line 12 in page 5, reference could be included for "classical Th1 genes")

Response: We apologise and have now added additional references to this particular section and throughout the manuscript.

- It would be helpful to specify accession numbers of RNA-sequencing datasets in the methods, similarly as is done for the ChIP-sequencing experiments (lines 16-17, page 14).

Response: We have added the accession numbers and details of the samples to new Table S1.

August 25, 2021

RE: Life Science Alliance Manuscript #LSA-2021-01075-TR

Prof. Richard G Jenner
University College London
UCL Cancer Institute
72 Huntley Street
London WC1E 6BT
United Kingdom

Dear Dr. Jenner,

Thank you for submitting your revised manuscript entitled "The Th1 cell regulatory circuitry is largely conserved between human and mouse". We would be happy to publish your paper in Life Science Alliance pending final revisions necessary to meet our formatting guidelines. Please also address Reviewer 2's remaining comment regarding Figure 2.

- please upload your main and supplementary figures as single files
- we encourage you to revise the figure legend for figure S2 such that the figure panels are introduced in an alphabetical order

LSA now encourages authors to provide a 30-60 second video where the study is briefly explained. We will use these videos on social media to promote the published paper and the presenting author. Corresponding or first-authors are welcome to submit the video. Please submit only one video per manuscript. The video can be emailed to contact@life-science-alliance.org

A. FINAL FILES:

-- High-resolution figure, supplementary figure and video files uploaded as individual files: See our detailed guidelines for preparing your production-ready images, <https://www.life-science->

alliance.org/authors

B. MANUSCRIPT ORGANIZATION AND FORMATTING:

Sincerely,

Reviewer #2 (Comments to the Authors (Required)):

In their revised manuscript, the authors have carefully addressed reviewer comments. This includes a number of additional analyses, which to the authors credit have been undertaken carefully, and interpreted with good balance and attention to confounding factors. The text of the manuscript has also been improved with regards to referencing and balance of the Discussion. The writing style is disciplined and very easy to follow.

On the whole, this is a much improved manuscript that comprehensively addresses the conservation and divergence in T-bet genomic binding and gene targets between human and mouse Th1 cells. The added discussion and analyses investigate in detail the underlying properties of conserved and single species T-bet binding events and target genes.

A minor comment the authors may want to address relates to the y-axis scales in Figure 2. In the current version, this scale is different for each category of binding events (columns in the figure), with the differences in coverage/binding intensity being substantial in some cases. This result is discussed in page 6 lines 20-23 of the manuscript. To facilitate independent interpretation of this result by the authors' readership, it would be helpful to use the same scale for each factor across all categories of binding events (i.e. same y-axis scale for each row in Figure 2).

September 7, 2021

RE: Life Science Alliance Manuscript #LSA-2021-01075-TRR

Prof. Richard G Jenner
University College London
UCL Cancer Institute
72 Huntley Street
London WC1E 6BT
United Kingdom

Dear Dr. Jenner,

Thank you for submitting your Research Article entitled "The Th1 cell regulatory circuitry is largely conserved between human and mouse". It is a pleasure to let you know that your manuscript is now accepted for publication in Life Science Alliance. Congratulations on this interesting work.

DISTRIBUTION OF MATERIALS:

Again, congratulations on a very nice paper. I hope you found the review process to be constructive and are pleased with how the manuscript was handled editorially. We look forward to future exciting submissions from your lab.

Sincerely,
